# Zinc-Dependent Oligomerization of *Thermus thermophilus* Trigger Factor Chaperone

**DOI:** 10.3390/biology10111106

**Published:** 2021-10-26

**Authors:** Haojie Zhu, Motonori Matsusaki, Taiga Sugawara, Koichiro Ishimori, Tomohide Saio

**Affiliations:** 1Institute of Advanced Medical Sciences, Tokushima University, Tokushima 770-8503, Japan; zhuhj@tokushima-u.ac.jp (H.Z.); matsusaki@tokushima-u.ac.jp (M.M.); 2Graduate School of Chemical Sciences and Engineering, Hokkaido University, Sapporo 060-8628, Japan; sugawara.taiga.p7@elms.hokudai.ac.jp; 3Department of Chemistry, Faculty of Science, Hokkaido University, Sapporo 060-0810, Japan

**Keywords:** trigger factor, zinc-dependent chaperone, *Thermus thermophilus*, thermal stability, secondary structure, mass spectrometry, oligomerization, NMR

## Abstract

**Simple Summary:**

Metal ions often play important roles in biological processes. *Thermus thermophilus* trigger factor (*Tt*TF) is a zinc-dependent molecular chaperone where Zn^2+^ has been shown to enhance its folding-arrest activity. However, the mechanisms of how Zn^2+^ binds to *Tt*TF and how Zn^2+^ affects the activity of *Tt*TF are yet to be elucidated. As a first step in understanding the mechanism, we performed in vitro biophysical experiments on *Tt*TF to investigate the zinc-binding site on *Tt*TF and unveil how Zn^2+^ alters the physical properties of *Tt*TF, including secondary structure, thermal stability, and oligomeric state. Our results showed that *Tt*TF binds Zn^2+^ in a 1:1 ratio, and all three domains of *Tt*TF are involved in zinc-binding. We found that Zn^2+^ does not affect the thermal stability of *Tt*TF, whereas it does induce partial structural change and promote the oligomerization of *Tt*TF. Given that the folding-arrest activity of *Escherichia coli* TF (*Ec*TF) is regulated by its oligomerization, our results imply that *Tt*TF exploits Zn^2+^ to modulate its oligomeric state to regulate the activity.

**Abstract:**

*Thermus thermophilus* trigger factor (*Tt*TF) is a zinc-dependent molecular chaperone whose folding-arrest activity is regulated by Zn^2+^. However, little is known about the mechanism of zinc-dependent regulation of the *Tt*TF activity. Here we exploit in vitro biophysical experiments to investigate zinc-binding, the oligomeric state, the secondary structure, and the thermal stability of *Tt*TF in the absence and presence of Zn^2+^. The data show that full-length *Tt*TF binds Zn^2+^, but the isolated domains and tandem domains of *Tt*TF do not bind to Zn^2+^. Furthermore, circular dichroism (CD) and nuclear magnetic resonance (NMR) spectra suggested that Zn^2+^-binding induces the partial structural changes of *Tt*TF, and size exclusion chromatography-multi-angle light scattering (SEC-MALS) showed that Zn^2+^ promotes *Tt*TF oligomerization. Given the previous work showing that the activity regulation of *E. coli* trigger factor is accompanied by oligomerization, the data suggest that *Tt*TF exploits zinc ions to induce the structural change coupled with the oligomerization to assemble the client-binding site, thereby effectively preventing proteins from misfolding in the thermal environment.

## 1. Introduction

In a crowded intracellular environment, newly synthesized polypeptide chains and metastable proteins risk misfolding or aggregation [1,2]. Molecular chaperones play the role of preventing proteins from misfolding or aggregation and removing the denatured proteins, thereby regulating the protein quality in the cell [3,4,5]. Chaperone-mediated protein quality control in bacteria has been extensively studied as a model system. One of the major molecular chaperones in bacterial cytosol, trigger factor (TF) chaperone, plays multiple roles in protein anti-aggregation, folding [6,7], translocation [8,9,10], and degradation [11]. TF binds to the ribosome to prevent the misfolding and aggregation of newly synthesized polypeptide chains [4,7,12,13,14,15]. For these versatile functions, TF has multiple activities, including “foldase activity”, which increases the folding rate and/or yield of the client proteins, and “holdase activity”, which halts or delays the folding of the client protein to prevent the misfolding of client proteins or to promote efficient protein translocation through Sec machinery [16,17]. Thus, TF seemingly has opposing activities in protein folding and switches the foldase/holdase activities depending on the circumstances.

One of the important aspects of TF for activity-switching is oligomerization [17]. It has been shown that *Escherichia coli* TF (*Ec*TF) forms a relatively weak dimer in the head-to-tail orientation [17,18] and that dimerization enhances holdase activity [17]. The assembly and rearrangement of the client-binding sites on TF induced by dimerization can modulate the binding kinetics with the client proteins, which explains the mechanism of the activity modulation [17].

Another strategy to modulate the activity of molecular chaperones is the binding of metal ions [19,20]. TF from *Thermus thermophilus* (HB8 strain) (*Tt*TF) is one of those chaperones, and it has been reported that the substrate folding-arrest activity (holdase activity) is activated by Zn^2+^ [20]. A previous study demonstrated that purified *Tt*TF binds Zn^2+^ and that zinc-binding can be saturated up to a 1:1 stoichiometric ratio by refolding in the presence of Zn^2+^. Although the previous study has shown the relationship between zinc-binding and holdase activity modulation [20], the mechanism of how Zn^2+^ alters the activity of *Tt*TF and whether it is related to oligomerization are unknown. Furthermore, *Tt*TF has no typical zinc-binding motif in its amino acid sequence, and thus the mechanism of how *Tt*TF recognizes Zn^2+^ ions is unclear.

Here, we focused on the zinc-dependent activity modulation of *Tt*TF and performed biophysical in vitro experiments, including the matrix-assisted laser desorption/ionization time-of-flight mass spectrometry (MALDI-TOF-MS), circular dichroism (CD), size exclusion chomatography-multi-angle light scattering (SEC-MALS), and solution nuclear magnetic resonance (NMR), to investigate the zinc-binding site of *Tt*TF, as well as zinc-dependent changes in the structure, thermal stability, and oligomeric state of *Tt*TF. The data show that all three domains of *Tt*TF are involved in the zinc-binding that induces partial structural changes and the oligomerization of *Tt*TF. Furthermore, given the relationship between the oligomerization and activity shown for *Ec*TF [17], our results suggest the mechanism of activity modulation of *Tt*TF, in which the Zn^2+^ alters structural properties to induce the oligomerization of *Tt*TF for activity regulation.

## 2. Materials and Methods

### 2.1. Plasmid Construction

The plasmids for protein expression in *E. coli* cells were constructed as follows. The synthetic gene fragments of *Tt*TF (1–404) and *Thermotoga maritima* TF (*Tm*TF) (1–425) were purchased from Thermo Fisher Scientific (Waltham, MA, USA) and inserted into the pCold vector after His_6_-tag (Takara, Kusatsu, Japan) to obtain the plasmid named pCold His_6_-*Tt*TF and pCold His_6_-*Tm*TF, respectively. The fragments and the vector were amplified by polymerase chain reaction using the primers summarized in Appendix A (Primers 1–8). The plasmid pCold His_6_-*Ec*TF was constructed in the previous study [13,17]. Given the potential binding of Zn^2+^ to His_6_-tag, the His_6_-tag needed to be removed for the zinc-binding assay. Therefore, pCold His_6_-TEV_CS_-*Tt*TF and pCold His_6_-TEV_CS_-*Ec*TF were constructed by inserting tobacco etch virus protease cleavage sites (TEV_CS_) into pCold His_6_-*Tt*TF and pCold His_6_-*Ec*TF, respectively. The amino acid residues sequences of TEV_CS_ for *Tt*TF and *Ec*TF are ENLYFQG and ENLYFQ, respectively. The primers used in these constructions are summarized in Appendix A (Primers 9–12).

The constructs of the isolated or tandem domains of *Tt*TF, including pCold His_6_-TEV_CS_-*Tt*TF^RBD^ (residues 1–113), pCold His_6_-TEV_CS_-*Tt*TF^SBD^ (residues 112–148 and 226–404, linked by GSGSG), pCold His_6_-TEV_CS_-*Tt*TF^PPD^ (residues 148–226), pCold His_6_-TEV_CS_-*Tt*TF^PPD-SBD^ (residues 112–404), and pCold His_6_-TEV_CS_-*Tt*TF^RBD-SBD^ (residues 1–148 and 226–404, linked by GSGSG) (Appendix A), were prepared using the primers and templates summarized in Appendix A (Primers 13–20).

The primers were purchased from Thermo Fisher Scientific and FASMAC (Kanagawa, Japan). The sequences of the constructed plasmids were verified by Eurofins Genomics (Tokyo, Japan).

### 2.2. Protein Expression and Purification

All protein samples used in this study were overexpressed in *E. coli* BL21 (DE3) cells and purified as described previously [11,13,20,21]. The sterilized Luria-Bertani (LB) medium (1 L) containing 50 mg/L ampicillin (Amp) was used to culture the cells at 37 °C. A total of 0.5 mM isopropyl β-D-1-thiogalactopyranoside (IPTG) was added to the medium when OD_600_ reached around 0.6–0.8. The cells were then cultured overnight at 18 °C. Cells were harvested by centrifugation at 4500 rpm for 15 min and resuspended in buffer containing 50 mM Tris-HCl and 0.5 M NaCl (pH 8.0). The cells were disrupted by a sonicator and centrifuged at 18,000 rpm for 45 min. The proteins were purified using Ni-NTA agarose (QIAGEN, Hilden, Germany). A total of 1 mg TEV protease was added to the protein from the 1 L medium to remove the His_6_-tag of *Ec*TF, followed by the incubation overnight at 4 °C. Similar operations were performed at room temperature to remove the His_6_-tag of *Tt*TF, but 2 mg TEV protease and longer digestion time (2 days) were required. After TEV protease digestion, the protein sample was incubated with Ni-NTA resin twice. The flowthrough was collected for further purification by gel filtration using a Superdex 200 16/600 column or Superdex 75 16/600 column (Cytiva, Marlborough, MA, USA). His_6_-tagged proteins (*Tm*TF, *Tt*TF^PPD-SBD^, and *Tt*TF^RBD-SBD^) and His_6_-tag removed proteins including *Ec*TF, *Tt*TF, and isolated domains of *Tt*TF (*Tt*TF^PPD^, *Tt*TF^SBD^, and *Tt*TF^RBD^) were used for the zinc-binding assay. His_6_-tagged *Tt*TF was used for CD, NMR, and SEC-MALS.

For the preparation of isotopically labeled *Tt*TF, *E. coli* BL21 (DE3) cells were cultured at 37 °C and in the 1 L sterilized M9 medium containing 6 g Na_2_HPO_4_, 3 g KH_2_PO_4_, 0.5 g NaCl, 2 g ^15^NH_4_Cl, 6 g D-glucose, 1.2 g MgSO_4_, 0.03 g thiamin (vitamin B_1_), and 50 mg Amp. [^13^CH_3_]-α-ketobutyric acid (50 mg L^−1^), [^13^CH_3_/^13^CH_3_]-α-ketoisovaleric acid (85 mg L^−1^), [^13^CH_3_]-methionine (50 mg L^−1^), and [2-^2^H, ^13^CH_3_]-alanine (50 mg L^−1^) were used to selectively label the methyl groups of Ile, Val, Leu, Met, and Ala residues, respectively, and these reagents were added to the M9 medium when OD_600_ reached around 0.5. When OD_600_ reached around 0.6–0.8, 0.5 mM IPTG was added to the M9 medium. Then, the cells were cultured overnight at 18 °C. The purification protocol of isotopically labeled proteins was the same as described above.

### 2.3. Preparation of Zinc-Bound and Zinc-Depleted Proteins

All purified proteins were diluted to below 2 μM and divided into two parts for the refolding operations as previously described [20]. One part was unfolded in the buffer containing 2 mM EDTA, 6 M guanidine, and 50 mM HEPES-KOH at pH 7.5 and at room temperature for 12 h. The other part was unfolded in the buffer containing 2 mM Zn(CH_3_COO)_2_, 6 M guanidine, and 50 mM HEPES-KOH at pH 7.5 for 12 h at room temperature. Then, proteins were refolded in the buffer containing 2 mM EDTA or 2 mM Zn(CH_3_COO)_2_ and 50 mM HEPES-KOH at pH 7.5 for 12 h at 4 °C. The extra EDTA or Zn(CH_3_COO)_2_ was removed by dialysis in buffer containing 50 mM HEPES-KOH at pH 7.5 and 4 °C for 6 h (twice). The refolded protein samples were used for in vitro biophysical experiments. Hereafter, TF proteins refolded in the Zn^2+^-containing and EDTA containing buffers are represented as TF (Zn^2+^) and TF (EDTA), respectively. The refolded proteins were evaluated by CD and SEC-MALS, which confirmed that the refolded proteins maintained the native structure (see Results).

### 2.4. MALDI-TOF-MS Spectrometry

For the mass spectrometry analysis, the protein samples, whose concentration was in the 20–100 μM range, were desalted by ZipTip (MILLIPORE, Bedford, MA, USA) and eluted by 1.5 μL saturated sinapinic acid solution (50% acetonitrile, 49.9% H_2_O, 0.1% TFA) on the target ground steel. After the protein sample was dried, the mass spectra were recorded by Microflex-TK mass spectrometer (Bruker Daltonics, Billerica, MA, USA) and Autoflex speed-DC mass spectrometer (Bruker Daltonics) in linear measurement mode. The protein standard I or II (Bruker Daltonics) were used for calibration.

The theoretical molar masses of *Tt*TF and its isolated domains after His_6_-tag removal are as below: 46,313 Da [full-length *Tt*TF (EDTA)], 12,706 Da [*Tt*TF^RBD^ (EDTA)], 25,618 Da [*Tt*TF^SBD^ (EDTA)], and 8911 Da [*Tt*TF^PPD^ (EDTA)]. The His_6_-tag of *Tm*TF (without the TEV cleaved site) was not removed, and the N-terminal of *Tm*TF still carried the MNHKVHHHHHH peptide fragment; thus, the theoretical molar mass of *Tm*TF (EDTA) is 51,330 Da. The theoretical molar mass of *Ec*TF (EDTA) after His_6_-tag removal is 48,193 Da. The His_6_-tag of *Tt*TF^PPD-SBD^ and GSGSG linked *Tt*TF^RBD-SBD^ is difficult to remove. Thus, the tandem domains with MNHKVHHHHHHENLYFQG peptide fragments were used in this experiment. The theoretical molar masses of *Tt*TF^PPD-SBD^ (EDTA) and GSGSG linked *Tt*TF^RBD-SBD^ (EDTA) are 36,193 Da and 40,249 Da, respectively.

### 2.5. Circular Dichroism Measurement

The CD spectra were recorded by a J-1500 CD spectrometer (JASCO, Tokyo, Japan). All protein samples were diluted to around 6 μM in a buffer containing 20 mM MOPS at pH 7.5. The temperature-dependent CD measurements were measured in the fixed wavelength of 222 nm from 20 °C to 95 °C at a 1 °C/min gradient. The CD spectra at the wavelength range from 190 nm to 320 nm were taken at fixed temperatures (20 °C, 35 °C, 50 °C, 65 °C, 80 °C, and 95 °C) 5 min after the cell temperature was stabilized. A quartz cell with 1 mm path length was used in all measurements. Protein concentration was determined based on the absorbance at 280 nm by UV-vis spectrometer JASCO V-730 (JASCO). The neural network program K2D [22] was used to analyze the secondary structure content of *Tt*TF (EDTA) and *Tt*TF (Zn^2+^) on Dichroweb web service (http://dichroweb.cryst.bbk.ac.uk/html/process.shtml, Accessed date: 14 September 2021 and 8–9 October 2021) [23]. The goodness-of-fit parameters (scaling factor) for the secondary structure analysis of *Tt*TF (EDTA) and *Tt*TF (Zn^2+^) were set to 0.95–1.05 [24].

### 2.6. Size Exclusion Chromatography in a Multi-Angle Laser Light Scattering (SEC-MALS)

SEC-MALS was measured using the instrument consisting of a high-performance liquid chromatography (HPLC) pump LC-20AD (Shimadzu, Kyoto, Japan), a refractive index detector RID-20A (Shimadzu), a UV-vis detector SPD-20A (Shimadzu), a light scattering detector DAWN HELEOS8+ (Wyatt Technology Corporation, Santa Barbara, CA, USA), and a TSKgel G3000SW_XL_ column (Tosoh Bioscience, Tokyo, Japan) [for *Tt*TF (EDTA) and *Tt*TF (Zn^2+^)] or KW-803 column (Shodex, Tokyo, Japan) (for natively purified *Tt*TF). The centrifuged (15,000 rpm for 5 min at 4 °C) *Tt*TF (Zn^2+^) or *Tt*TF (EDTA) samples at various concentrations in buffer containing 50 mM HEPES-KOH and 100 mM KCl at a pH of 7.5, were injected into the HPLC system at the 1 mL/min flow rate. Data were analyzed with ASTRA version 7 (Wyatt Technology Corporation).

### 2.7. NMR Spectroscopy

Both *Tt*TF (EDTA) and *Tt*TF (Zn^2+^) were concentrated to ~0.1 mM and prepared in a buffer containing 50 mM HEPES-KOH (pH 7.5) and 100 mM KCl. ^1^H-^13^C heteronuclear multiple quantum coherence (HMQC) spectra were recorded on Bruker Avance III 500 MHz NMR (Bruker Biospin AG, Fällanden, Switzerland) at 65 °C. The spectra were processed with the NMRPipe software [25]. Olivia software (https://github.com/yokochi47/Olivia, Accessed date: 22 December 2017) was used to analyze the spectral data.

## 3. Results

### 3.1. Zinc-Binding Is Characteristic to TtTF

It has previously been shown that Zn^2+^ binds to *Tt*TF in a 1:1 stoichiometric ratio [20]. To investigate whether the Zn^2+^-binding is a unique character of *Tt*TF, TF chaperones from three different organisms, *Thermus thermophilus*, *Thermotoga maritima*, and *Escherichia coli* (Appendix A), were subjected to the zinc-binding assay. Following the procedure described in the previous report [20], the purified TF chaperones, *Tt*TF, *Tm*TF, and *Ec*TF, were refolded in the presence of Zn(CH_3_COO)_2_ or EDTA. After buffer exchange by dialysis, the proteins were analyzed by MALDI-TOF-MS, which is widely used in metalloproteomics [26,27]. The molar mass of *Tt*TF refolded in the presence of Zn(CH_3_COO)_2_ [*Tt*TF (Zn^2+^)] was 62 ± 12 Da larger than *Tt*TF refolded in the presence of EDTA [*Tt*TF (EDTA)] (Figure 1A, Table 1). Given the molar mass of Zn^2+^ (65 Da), the data showed that *Tt*TF binds Zn^2+^ in a stoichiometric ratio of 1:1, which is consistent with the previous report [20]. Note that the previous study on *Tt*TF exploiting inductively coupled plasma spectroscopy (ICPS) element analysis and spectroscopic titration experiment identified that the purified *Tt*TF from *E. coli* cells contains a half equimolar Zn^2+^, which can be saturated at a 1:1 stoichiometric ratio by zinc-saturation by refolding [20]. These data support the idea that the 1:1 zinc-binding seen in the refolded *Tt*TF represents the functional binding. Conversely, *Ec*TF and *Tm*TF showed negligible molar mass differences derived from Zn^2+^ treatment (Figure 1B,C, Table 1). Therefore, the data showed that *Tm*TF and *Ec*TF bind no Zn^2+^, and Zn^2+^-binding is characteristic to *Tt*TF.

### 3.2. Full-Length of TtTF Was Required for Zinc Recognition

We next investigated the zinc-binding sites on *Tt*TF. Although *Tt*TF has been shown to bind Zn^2+^ at a 1:1 stoichiometry (Figure 1A), the amino acid sequence of *Tt*TF does not have a typical zinc-binding motif. To investigate the zinc-binding sites on *Tt*TF, *Tt*TF was divided into the three domains: the ribosome-binding domain (*Tt*TF^RBD^: Residues 1 to 113), the substrate-binding domain (*Tt*TF^SBD^: Residues 112 to 148 and 226 to 404, connected by the GSGSG linker), and the peptidyl-prolyl isomerase domain (*Tt*TF^PPD^: Residues 148 to 226) (Figure 2A and Appendix A). The domain boundaries of *Tt*TF were defined according to the sequence alignment with *Ec*TF, the structures of *Ec*TF [12,13,17], and the Alphafold2 predicted structural model of *Tt*TF [28] (Appendix A). The zinc-binding assay for the isolated domains was performed by following the same procedure as for the full-length TFs. Each of the domains was refolded in the presence of Zn(CH_3_COO)_2_ or EDTA and subjected to MALDI-TOF-MS analysis (Figure 2B–D, Table 1). As summarized in Table 1, all three domains indicated that the zinc-dependent mass difference is negligible compared to the mass of Zn^2+^ (65 Da). Thus, no zinc-binding was detected for the isolated domains.

Zinc-binding was also tested for the tandem domains of *Tt*TF, *Tt*TF^PPD-SBD^, and *Tt*TF^RBD-SBD^ (Figure 2E,F). The data showed that the zinc-dependent mass difference is negligible compared to the mass of Zn^2+^ (65 Da). Thus, no zinc-binding was detected for the tandem domains. Note that the larger differences between the experimental and the theoretical masses for tandem domains (−188 Da for *Tt*TF^PPD-SBD^ and −151 Da for *Tt*TF^RBD-SBD^) can be explained by the removal of the N-terminal methionine residue (149 Da) by the methionyl-aminopeptidase during protein expression in *E. coli* [29]. Collectively, our data indicated that neither a single domain nor a tandem domain binds Zn^2+^. In contrast, the full-length of *Tt*TF is required for zinc recognition, implying that all three domains are involved in zinc recognition.

### 3.3. Zn^2+^ Induced Little Effect to Thermal Stability of TtTF

One of the possible benefits of binding metal ions is the improvement in thermal stability. To test if Zn^2+^ influences the thermal stability of *Tt*TF, the mean residue ellipticity at 222 nm, indicative of α-helix, was monitored for *Tt*TF (Zn^2+^) and *Tt*TF (EDTA) at an increasing temperature from 20 °C to 95 °C (Figure 3A). At the range of 20–80 °C, *Tt*TF (Zn^2+^) showed a smaller magnitude of the mean residue ellipticities compared to *Tt*TF (EDTA) (Figure 3A). In both cases, the mean residue ellipticity value gradually increased with the temperature in the range of 20–80 °C, whereas the slope became steeper above 85 °C, indicating that the protein started unfolding above 85 °C. This observation is consistent with the fact that the maximum growth temperature of *Thermus thermophilus* is 85 °C [30]. Due to the limited temperature range of the CD measurement, the unfolding was incomplete, and accordingly, the midpoint of the unfolding transition temperature *T*_m_ and thermodynamic parameters [31,32] could not be determined. However, the similar melting profiles of *Tt*TF (Zn^2+^) and *Tt*TF (EDTA) in the temperature range of 85–95 °C suggested that zinc-binding had little effect on the thermal stability of *Tt*TF in this temperature range.

### 3.4. Zn^2+^ Induced Partial Structural Change of TtTF

Although Zn^2+^ binding has little effect on the thermal stability of *Tt*TF, the data showed that *Tt*TF (Zn^2+^) has a weaker ellipticity at 222 nm than *Tt*TF (EDTA) (Figure 3A), suggesting the change in the secondary structure of *Tt*TF upon binding to Zn^2+^. CD spectra were acquired at 50 °C (Figure 3B) to investigate the effect of zinc-binding to the secondary structure of *Tt*TF. Compared to *Tt*TF (EDTA), the CD spectrum of *Tt*TF (Zn^2+^) showed a smaller magnitude of the mean residue ellipticities for the region from 208 nm to 222 nm (the negative absorption peak of α-helix), indicating that *Tt*TF (Zn^2+^) contains less α-helix. The secondary structure content predicted from the CD spectra using Dichroweb [23] showed that *Tt*TF (Zn^2+^) contains a smaller portion of α-helix but contains a larger portion of β-sheet (Figure 3B, Table 2), indicating the zinc-induced structural changes of *Tt*TF. CD measurements at different temperature points, 20 °C, 35 °C, 65 °C, and 80 °C, showed essentially the same trend, that the mean residual ellipticity of *Tt*TF (Zn^2+^) is less negative than that of *Tt*TF (EDTA) (Appendix A and Table 2). Note that the purified *Tt*TF before refolding showed essentially the same CD spectrum as *Tt*TF (EDTA), supporting the idea that the *Tt*TF preserves its native structure even after the refolding process (Appendix A).

The effect of zinc-binding was further investigated by NMR, in which the methyl-HMQC spectra for [U-^15^N; Ala-^13^CH_3_; Met-^13^CH_3_; Ile-δ1-^13^CH_3_; Leu/Val-^13^CH_3_/^13^CH_3_]-labeled *Tt*TF (Zn^2+^), and *Tt*TF (EDTA) were acquired (Figure 3C). The spectrum of *Tt*TF (EDTA) showed well-dispersed methyl resonances, whereas that of *Tt*TF (Zn^2+^) showed fewer resonances, and several resonances were missing, most probably due to line broadening. The line broadening of *Tt*TF (Zn^2+^) can be explained by exchange broadening due to the conformational exchange in a μs-ms timescale or by an increase in size due to oligomerization.

### 3.5. Zn^2+^ Promoted Oligomerization of TtTF

From the above results, we found that Zn^2+^ induces the partial structural changes of *Tt*TF. SEC-MALS experiments were performed to test if the partial structural change induced by zinc-binding affects the oligomeric state of *Tt*TF, which can be associated with the holdase activity of *Tt*TF. Although the previous study reported that *Tt*TF exists as a monomer at a lower concentration [20], the oligomeric state of *Tt*TF at higher concentrations has not been investigated. Furthermore, TF from other organisms, including *E. coli.* and *V. cholerae*, have been shown to form a dimer at higher concentrations [33,34]. The dissociation constant (*Kd*) of *Ec*TF monomer–dimer equilibrium in solution was estimated as ~2 µM [17], and the crystal structure of *V. cholerae* TF was solved as a dimer [34]. The SEC-MALS data at lower concentrations showed that *Tt*TF exists as a monomer (Figure 4), which is consistent with the previous gel-filtration analysis [20]. In contrast, the observed molar mass increased at higher concentrations, indicating that *Tt*TF undergoes concentration-dependent oligomerization (Figure 4). Note that the molar mass observed by SEC-MALS reflects the average molar mass of the protein if the protein exists in equilibrium among multiple oligomeric states. Interestingly, *Tt*TF (Zn^2+^) oligomerized at a lower concentration compared to *Tt*TF (EDTA) and the purified *Tt*TF (Figure 4C and Appendix A). For example, the average molar mass for *Tt*TF (Zn^2+^) at ~14 μM was estimated as ~76 kDa that is close to the theoretical molar mass for the dimer (92 kDa), whereas the average molar mass for *Tt*TF (EDTA) and the natively purified *Tt*TF at ~20 μM were both estimated as 59 kDa (Figure 4C and Appendix A). Thus, the data indicate that zinc-binding promoted the oligomerization of *Tt*TF.

## 4. Discussion

*Tt*TF has a unique character among other homologous TFs: it binds the zinc ion with a 1:1 stoichiometry, and zinc-binding regulates the holdase activity of *Tt*TF [20]. However, the mechanism of how Zn^2+^ regulates *Tt*TF activity has been poorly understood. In general, one of the possible benefits of the metal ion is to improve the thermal stability of proteins [35,36,37], but our data showed no significant thermal stability change between *Tt*TF (Zn^2+^) and *Tt*TF (EDTA) (Figure 3A), and thus this is not the case with *Tt*TF. Interestingly, our SEC-MALS data show Zn^2+^ promotes the oligomerization of *Tt*TF (Figure 4), which can be one of the key features in explaining the zinc-dependent activity modulation of *Tt*TF. A previous study on *Ec*TF showed that oligomerization promotes holdase activity, in which the dimer formation of *Ec*TF assembles the client-binding sites and thus can accelerate the association kinetics with the client proteins [17]. Faster association kinetics have been shown to enhance the holdase activity of chaperones [17]. Thus, the zinc-dependent oligomerization of *Tt*TF suggests that *Tt*TF exploits Zn^2+^ to modulate the binding kinetics and thus activity through oligomerization (Figure 5).

In addition to oligomerization, zinc-binding induces the partial structural changes of *Tt*TF, as shown by CD (Figure 3B, Appendix A, and Table 2) and NMR (Figure 3C). The NMR analysis of *Tt*TF in the absence and presence of Zn^2+^ indicated that several hydrophobic amino acids, including Leu and Val, are affected by zinc-binding, which thus implies that the hydrophobic region of *Tt*TF is involved in the partial structural changes (Figure 3C). Previous structural studies on *Ec*TF show that both client-binding sites and the dimer-interface consist mainly of hydrophobic residues. The regions undergoing zinc-induced structural change can be a part of the oligomeric interface and/or a part of the client-binding site (Figure 5).

Collectively, our data from biophysical experiments suggest that Zn^2+^ binds to induce structural change and oligomerization of *Tt*TF, which can be important features to activate the holdase activity of *Tt*TF. Although further structural research is needed, our study provides mechanistic insight into the zinc-dependent activity modulation of *Tt*TF.

## 5. Conclusions

A series of biophysical experiments were performed to investigate the physical properties of Zn^2+^-dependent *Tt*TF. Although the specific binding sites of Zn^2+^ have not been identified in this study, MALDI-TOF-MS experiments show that the full-length of *Tt*TF is involved in zinc coordination. CD, NMR, and SEC-MALS showed that zinc-binding induces the partial structural changes of *Tt*TF and promotes the oligomerization of *Tt*TF. Given the previous report on *Ec*TF showing the relationship between the oligomerization and the activity modulation [17], the zinc-dependent oligomerization of *Tt*TF can be one of the key features to modulate the activity of *Tt*TF.

## Figures and Tables

**Figure 1 biology-10-01106-f001:**
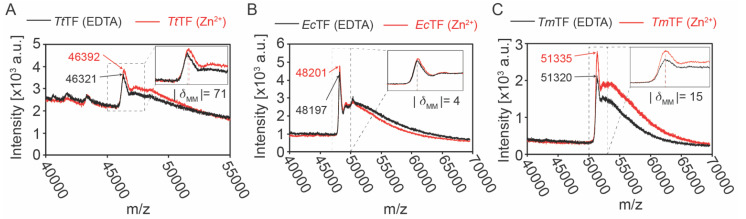
MALDI-TOF-MS analysis for *Tt*TF, *Tm*TF, and *Ec*TF prepared in the absence and presence of Zn^2+^. The overlapped mass spectra for *Tt*TF (**A**), *Ec*TF (**B**), and *Tm*TF (**C**) refolded in the buffer containing Zn^2+^ (red) or EDTA (black). The expanded views are displayed at the top right corner. The theoretical molar masses of *Tt*TF (EDTA), *Ec*TF (EDTA), and *Tm*TF (EDTA) are 46,313 Da, 48,193 Da, and 51,330 Da, respectively. The theoretical molar mass of Zn^2+^ is 65 Da. The absolute values of the difference between the experimental molar masses (|*δ*_MM_|) of TFs (Zn^2+^) and TFs (EDTA) are displayed in the below expanded views. a.u., arbitrary unit.

**Figure 2 biology-10-01106-f002:**
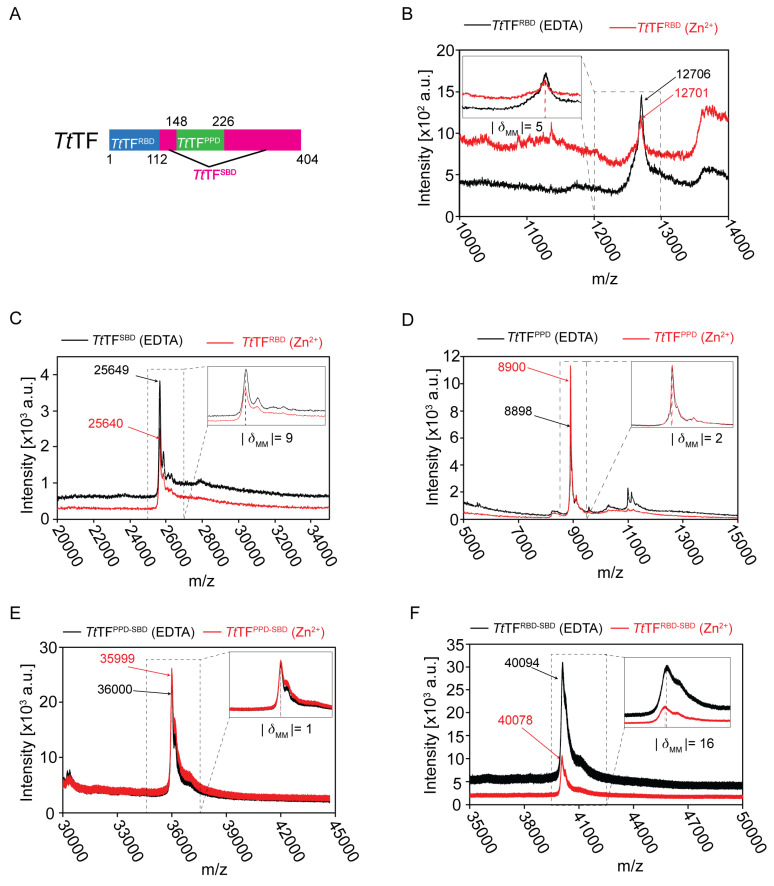
MALDI-TOF-MS analysis for the *Tt*TF domains prepared in the absence and presence of Zn^2+^. (**A**) The domain architecture of *Tt*TF. (**B**–**F**) The overlaid mass spectra for *Tt*TF^RBD^ (**B**), *Tt*TF^SBD^ (**C**), *Tt*TF^PPD^ (**D**), *Tt*TF^PPD-SBD^ (**E**), and *Tt*TF^RBD-SBD^ (**F**) refolded in the presence of Zn^2+^ (red) and EDTA (black). The expanded views are displayed at the top right or top left. The theoretical molar mass: 12,706 Da for *Tt*TF^RBD^ (EDTA), 25,618 Da for *Tt*TF^SBD^ (EDTA), 8911 Da for *Tt*TF^PPD^ (EDTA), 36,193 Da for *Tt*TF^PPD-SBD^ (EDTA), and 40,249 Da for *Tt*TF^RBD-SBD^ (EDTA). The absolute values of the difference between the experimental molar masses (|*δ*_MM_|) of *Tt*TF^domains^ (Zn^2+^) and *Tt*TF^domains^ (EDTA) are shown below the expanded views. a.u., arbitrary unit.

**Figure 3 biology-10-01106-f003:**
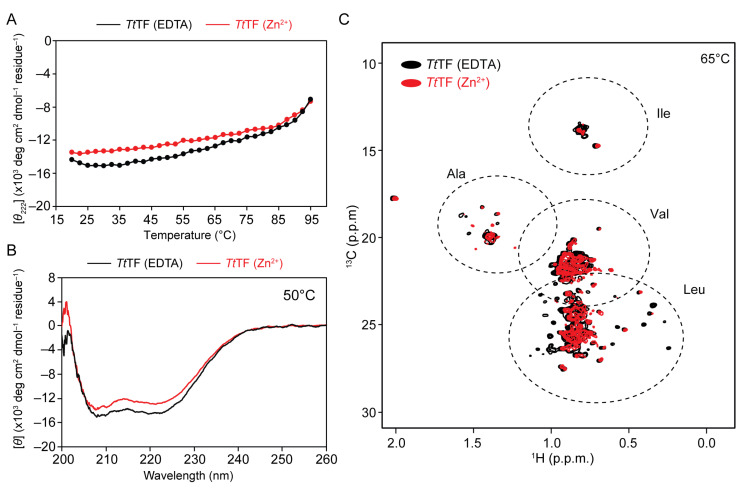
Structural characterization of *Tt*TF in the absence and presence of Zn^2+^. (**A**) Temperature scan of CD ellipticity at 222 nm performed from 20 °C to 95 °C. The plots for *Tt*TF (Zn^2+^) and *Tt*TF (EDTA) are shown in red and black, respectively. (**B**) CD spectra of *Tt*TF (Zn^2+^) (red) and *Tt*TF (EDTA) (black) at 50 °C. (**C**) The HMQC spectra of [U-^15^N; Ala-^13^CH_3_; Met-^13^CH_3_; Ile-δ1-^13^CH_3_; Leu/Val-^13^CH_3_/^13^CH_3_]-labeled *Tt*TF (Zn^2+^) (red) and *Tt*TF (EDTA) (black). Typical regions for the methyl resonances from Ala, Ile δ1, Val, and Lue are indicated by dashed eclipses. The NMR experiments were performed at 65 °C.

**Figure 4 biology-10-01106-f004:**
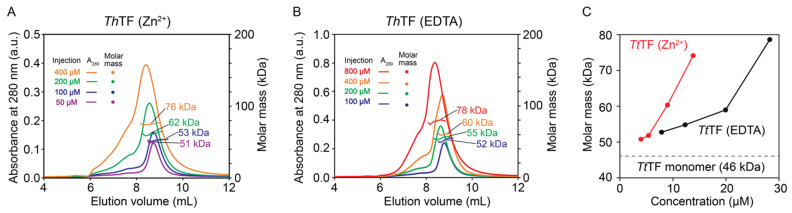
SEC-MALS for the investigation of the oligomeric state of *Tt*TF. (**A**,**B**) SEC-MALS profiles of *Tt*TF (Zn^2+^) (**A**) and *Tt*TF (EDTA) (**B**) injected at varying concentrations. At higher concentration, the larger molar mass was observed for *Tt*TF (Zn^2+^) and *Tt*TF (EDTA), indicating the increase of the oligomeric fractions at higher concentrations. (**C**) Plots of the molar masses of *Tt*TF (Zn^2+^) (red) and *Tt*TF (EDTA) (black) with respect to the concentrations measured by the refractive index at the peak top. Note that *Tt*TF is diluted after injection into the SEC column, and thus, the concentration at the peak top is lower than that at the injection. a.u., arbitrary unit.

**Figure 5 biology-10-01106-f005:**
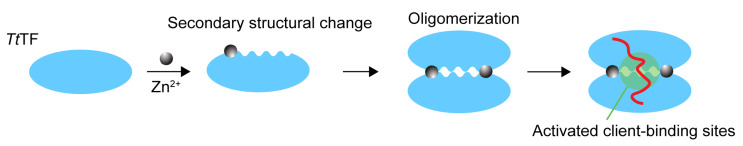
Schematic representation of a possible mechanism for the zinc-dependent activity modulation of *Tt*TF. Zn^2+^ induces the partial structural changes and promotes the oligomerization of *Tt*TF. Oligomerization can assemble the client-binding sites that enables the efficient client binding, as seen in the dimeric *Ec*TF. The client protein is represented as a red line.

**Table 1 biology-10-01106-t001:** Summary of MALDI-TOF-MS experimental results for TtTF and its domains in the presence/absence of Zn^2+^. The values are rounded to integers.

	Ave ± SD (Da) (Refolded in Zn^2+^ Containing Buffer)	Ave ± SD (Da) (Refolded in EDTA Containing Buffer)	δ (Da)
*Tt*TF	46381 ± 8	46319 ± 4	62 ± 12
*Tm*TF	51338 ± 13	51335 ± 12	3 ± 25
*Ec*TF	48218 ± 12	48214 ± 11	4 ± 22
*Tt*TF^RBD^	12698 ± 2	12702 ± 3	−4 ± 5
*Tt*TF^SBD^	25646 ± 5	25648 ± 2	−2 ± 6
*Tt*TF^PPD^	8899 ± 3	8899 ± 2	0 ± 4
*Tt*TF^PPD-SBD^	35990 ± 6	36005 ± 4	−15 ± 10
*Tt*TF^RBD-SBD^	40075 ± 4	40098 ± 6	−23 ± 9

**Table 2 biology-10-01106-t002:** Summary of the secondary structure contents of *Tt*TF in the absence and presence of Zn^2+^ at 20 °C, 35 °C, 50 °C, and 65 °C. The fitting failed for the data of *Tt*TF (Zn^2+^) at 20 °C and *Tt*TF (EDTA) at 65 °C.

Temperature (°C)	State of *Tt*TF	α-Helix	β-Sheet	Random Coil
20	*Tt*TF (EDTA)	56%	9%	35%
	*Tt*TF (Zn^2+^)	-	-	-
35	*Tt*TF (EDTA)	59%	8%	34%
	*Tt*TF (Zn^2+^)	47%	18%	35%
50	*Tt*TF (EDTA)	54%	10%	36%
	*Tt*TF (Zn^2+^)	46%	17%	37%
65	*Tt*TF (EDTA)	-	-	-
	*Tt*TF (Zn^2+^)	42%	20%	37%

## Data Availability

The authors declare that all data supporting the findings of this study are available within the paper. All other information is available from the corresponding authors upon reasonable request.

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
