# Peer review of "Zinc-Dependent Oligomerization of Thermus thermophilus Trigger Factor Chaperone"

_biology, 2021, doi:10.3390/biology10111106_

Round 1
Reviewer 1 Report
Please see attachment.

Reviewer 2 Report
The manuscript "Zinc-dependent oligomerization of Thermus thermophilus trigger factor chaperone" (biology-1415786) by Zhu et al. Moreover, the current form of manuscript requires some modifications/clarifications and citing new references need to be addressed prior publication in biomolecules.
- Author should calculate the thermodynamic parameters. There is no meaning to show a simple plot of temperature dependent (figure 3A) spectral change.
- There is no table for data summary for figure 3 A and B. There is a lot of data that needs to be summarized in the table.
- Deconvoluted spectra analysis of far-UV CD spectra should perform to elucidate the deep secondary structure change.
- Page 2, line 45-46: “In a crowded intracellular environment, newly synthesized polypeptide chains and metastable proteins risk misfolding or aggregation”. For that, author took help from available literature such as and cite these article: Int J Biol Macromol. 2018, 109:483-491.
- Page 2, line 59: “One of the important aspects of TF for activity switching is oligomerization”. Sentence is not supported with literature.
Reviewer 3 Report
This manuscript describes biophysical experiments characterizing the role of zinc binding in Thermus thermophilus trigger factor. MALDI TOF suggests each trigger factor monomer binds one zinc ion, CD and NMR suggest a structural transition upon zinc binding and SEC-MALS suggests that zinc binding triggers dimerization. This is an interesting system and the results enhance the available knowledge base. A major shortcoming is that the mode of zinc binding was not investigated.
COMMENTS:
- In the introduction or section 3.1, briefly define and comment on the degree of sequence conservation between the trigger factor sequences of the three species studied in these experiments. A sequence alignment in supplementary materials would be useful. Also, briefly define the domains of trigger factor and their function in the introduction or section 3.2. State how the individual domain boundaries were defined, based on sequence or structure conservation?
- Table 1 has some inconsistencies. For the last two constructs, the difference in mass should not be negative. Also, for these two dual domain constructs, please explain the large differences between the observed and theoretical mass (-188 and -151).
- In general and particularly for intact ThTF and its dual domain constructs, do you see oligomers in the MALDI-TOF data?
- The CD results in Fig 3 and S2 suggest a structural transition with small decrease in alpha helix and increase in beta sheet, not partial unfolding as stated in the legend of Fig 5. Previous work (reference 17) stated that there was no difference in the CD spectra of ThTF with and without zinc; please explain the possible difference. At 6uM CD concentration, will the Zn+ and Zn- forms both be monomeric? Do you see concentration dependent effects in the CD spectra?
- In section 3.5, please be more specific about the protein concentrations discussed for monomer-dimer transitions (not lower and higher) in the TF proteins all the various species discussed.
- Previous studies (Morgado 1992 Nature Communications 8; The dynamic dimer structure of the chaperone Trigger Factor) of e coli TF showed that having a His-tag on the protein can affect the SEC-MALS profile, with the His-tag enhancing oligomerization. Did you test ThTF with and without His tag in this assay?
- Can you discuss potential mode of zinc binding, eg what residues could be involved? Are these residues conserved in TF from any other organisms?
- Minor comments: Figure 1b,c labels should refer to the correct organisms. Figure 1 and others, typo arbitrary.
Reviewer 4 Report
The paper of Zhu et al. deals with the effect of zinc ion on ThTF chaperonstability and oligomerization.
In my opinion the work deserves publication in Biology after some revision.
- The domain organization of ThTF (figure S1) should be included in the text,
to make the reading clearer
- The statement that "full-length of ThTF is required for zinc recognition,
implying that all three domains are involved in zinc recognition."
(line 233-234) may be rewritten based on the model in Figure 5.
Alternatively, this sentence can be removed since it suggests that ThTF
assumes a tertiary structure with a specific zinc pocket in the monomeric
state. Since this is not the case, based on their experiments, this
sentency must be rewritten or removed.
- Figure S3 should be moved in the main text since it relates to
experimental results in that section.
Round 2
Reviewer 1 Report
In their response letter the authors have sufficiently addressed my previous concerns. The added clarifications and explanations facilitate reading and understanding of the manuscript. I therefore support its publication.